# PAAD: Panelization algorithm for architectural designs

Andrew Fisher[1], Xing Tan[2], Muntasir Billah[3], Pawan Lingras[1], Jimmy Huang[4], Vijay Mago[5]*

1 Department of Mathematics and Computing Science, Saint Mary's University, Halifax, Nova Scotia, Canada, 2 Department of Computer Science, Lakehead University, Thunder Bay, Ontario, Canada, 3 Department of Civil Engineering, University of Calgary, Calgary, Alberta, Canada, 4 School of Information Technology, York University, Toronto, Ontario, Canada, 5 School of Health Policy and Management, York University, Toronto, Ontario, Canada

* vmago@yorku.ca

**Data Availability Statement:** The research was based on the analysis of geometric data that can be publicly accessed through our Github repository (https://github.com/andrfish/PAAD) in the "Data. java" file.

## Abstract

Due to the competitive nature of the construction industry, the efficiency of requirement analysis is important in enhancing client satisfaction and a company's reputation. For example, determining the optimal configuration of panels (generally called panelization) that form the structure of a building is one aspect of cost estimation. However, existing methods typically rely on rule-based approaches that may lead to suboptimal material usage, particularly in complex designs featuring angled walls and openings. Such inefficiency can increase costs and environmental impact due to unnecessary material waste. To address these challenges, this research proposes a Panelization Algorithm for Architectural Designs, referred to as PAAD, which utilizes a genetic evolutionary strategy built on the 2D bin packing problem. This method is designed to balance between strict adherence to manufacturing constraints and the objective of optimizing material usage. PAAD starts with multiple potential solutions within the predefined problem space, facilitating dynamic exploration of panel configurations. It approaches structural rules as flexible constraints, making necessary corrections in post-processing, and through iterative developments, the algorithm refines panel sets to minimize material use. The methodology is validated through an analysis against an industry implementation and expert-derived solutions, highlighting PAAD's ability to surpass existing results and reduce the need for manual corrections. Additionally, to motivate future research, a synthetic data generator, the architectural drawing encodings used, and a preliminary interface are also introduced. This not only highlights the algorithm's practical applicability but also encourages its use in real-world scenarios.

## Introduction

In the domain of construction project management, the development of manufacturing plans is an important stage. This involves optimizing the material utilization [1], which can impact several key aspects of the project such as timeline, budget, and generated waste [2, 3]. This

**Funding:** MITACS - IT22468. The funders had no role in study design, data collection and analysis, decision to publish, or preparation of the manuscript.

**Competing interests:** The authors have declared that no competing interests exist.

efficiency, in turn, can affect client satisfaction as well as the reputation of the construction firm, which is important in a competitive market. When a manufacturing plan is suboptimal, it can lead to negative consequences such as loss in business opportunities [4], as well as delays in subsequent projects or reduced profit margins due to unnecessary material wastage. For example, consider the *panelization* process which is often used to develop building structures in a controlled factory environment [3, 5]. This procedure involves manufacturing numerous *panels* or rectangular frames that collectively form different parts of a building, such as walls, floors, or ceilings [6]. However, a significant challenge in doing this is ensuring that the panels avoid obstructing openings such as windows or doors, and accommodating angled walls. This introduces added complexities to the task, making it difficult to determine the most optimal solution in a timely manner [7].

To address this challenge, research has focused on automating the panelization process through the use of algorithmic approaches [3, 5, 7]. Collaboration with an industry partner revealed that real-world applications often apply a rule-based method to ensure structural integrity and compliance with manufacturing constraints. This approach can be difficult to maintain as it is required to perform numerous checks and exceptions in order to adhere to such rules. Additionally, if an outlier case is introduced, a new set of rules would need to be defined in order to accommodate it. While this approach ensures that the output will always be feasible for manufacturing, it can result in solutions that are suboptimal in terms of material efficiency. This is due to the process overfitting each rule, potentially leading to excessive material use or constraints in the design's flexibility.

To develop a solution that balances adherence to industry rules with optimal material usage, the implementation of a genetic evolutionary algorithm [8] is proposed called PAAD: Panelization Algorithm for Architectural Designs. This method works by initializing multiple possible solutions or populations within the problem space (i.e., a pre-defined wall), each exploring different combinations and configurations of panels. It treats manufacturing rules as hard constraints and structural rules as flexible ones, by applying post-processing techniques to ensure compliance. Through several iterations or evolutions, the approach refines panel sets for increased optimality by reducing material wastage. This is achieved by building on the 2D bin packing (2DBP) problem [9], where PAAD aims to efficiently generate the minimal number of panels needed to completely cover a given wall. The most optimal solution is then identified at the end of each evolution, with modifications made through the use of mutation and crossover operators [8].

The evolutionary process continues for several iterations, investigating how these solutions could be further optimized until the results are deemed acceptable for manufacturing and/or PAAD is unable to optimize it further. This approach is able to handle complex walls that contain openings as well as angled lines, which can be observed in the high-level overview depicted in Fig 1. As denoted by the black squares, "void space" is used to accommodate these complexities by being interpreted as stationary objects that cannot be moved. This forces the algorithm to pack around these areas and minimize overall wastage since they shouldn't be considered in the final output. Consequently, PAAD represents a dynamic and adaptable strategy [10, 11] that consistently refines and evolves to identify the most material-efficient solution that is compliant with industry standards. To the best of our knowledge, this collaborative effort with the industry resulted in the following contributions:

- A novel genetic evolutionary approach for the panelization problem that complies with manufacturing constraints

- A generalizable framework that can be adapted to different structural constraints as well as building definitions

**Step 1:** Define the problem space

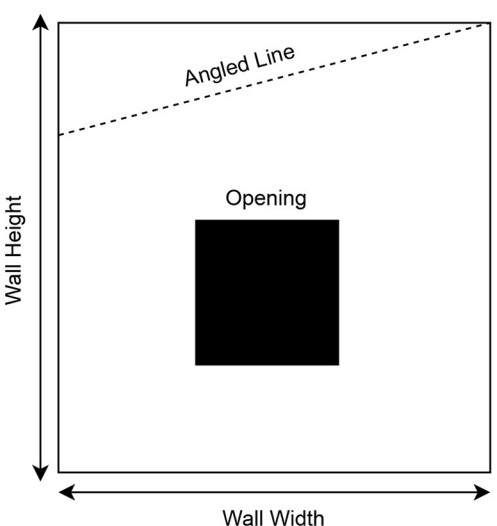

**Step 2:** Generate a set of panelization solutions

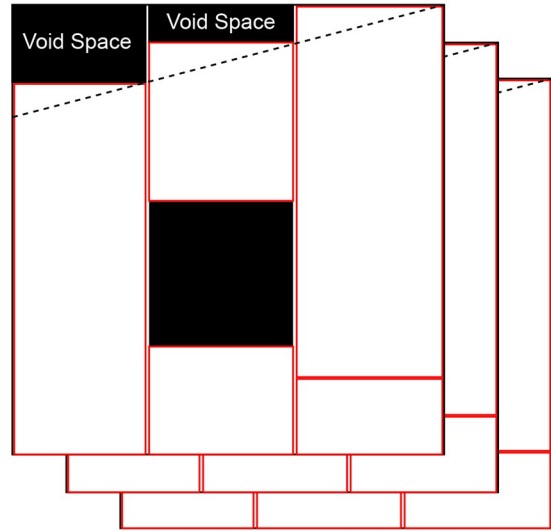

**Step 3:** Perform mutation and crossover operations, then select the fittest solution

**Step 4:** Generate a new set of solutions that are derived from the fittest solution

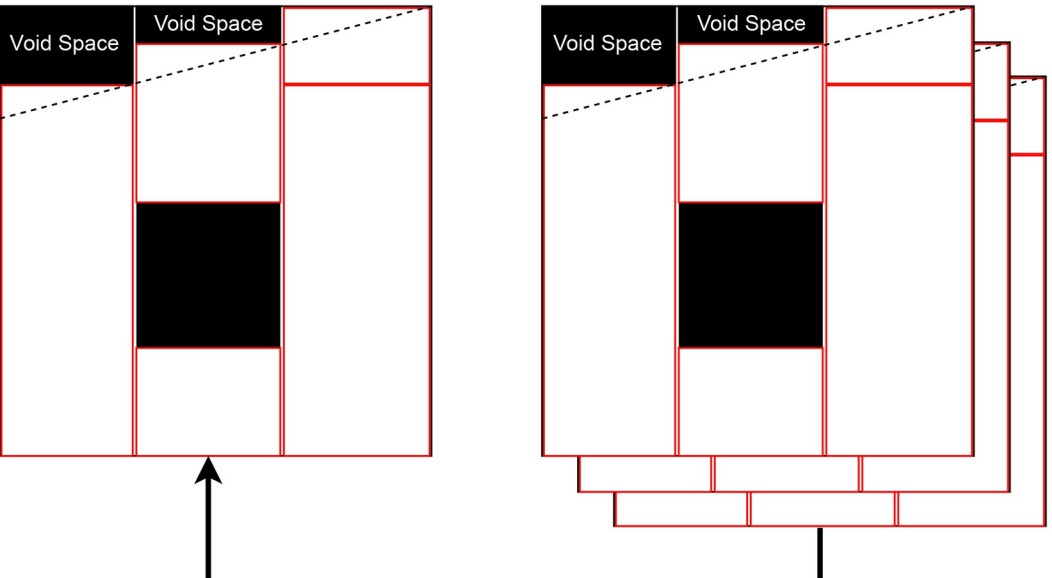

**Step 5:** Repeat the process until unable to exit a local minima or maximum number of evolutions has been reached

**Fig 1. Overview.** A high-level overview of the process PAAD follows, which is based on the 2D bin packing problem and uses a genetic evolutionary approach.

- A comparison with industry experts and current industry standards to demonstrate the advantages of the proposed algorithm

- A synthetic data generator as well as architectural drawing encodings to overcome the confidentiality issues associated with sharing building plan imagery

The article begins with a review of related works addressing both panelization and the 2D bin packing problem. Next, the dataset structure is described and a data repository is introduced, designed to facilitate future research. The methodology section then describes on how the proposed PAAD framework automates panelization, considering manufacturing constraints and using post-processing to accommodate structural constraints. Lastly, the results section presents a comparative analysis of the improvements over a current industry implementation alongside manually labeled data from domain experts.

## Related works

In the area of construction panelization, recent advancements have used algorithmic methods to tackle the challenges associated with material optimization [12]. The integration of Building Information Modeling (BIM) and domain-specific adaptations has been important in applying these advancements to real-world scenarios. For instance, BIM-based generative frameworks have been developed to automate the design of offsite construction components, such as panels [13, 14]. These frameworks apply a generative design algorithm that automates the panelization process by utilizing rule-based methods and discrete event simulation models. The former generates potential solutions based on BIM attributes, while the latter assesses their practicality [13].

Similarily, BIM has been used in algorithms for panelizing residential building drainage systems, by applying rule-based approaches to automate the design process [3]. These algorithms apply a series of predetermined rules to dictate the layout and specifications of drainage systems, with the goal of enhancing efficiency and reducing material waste. In particular, Integer Programming (IP) has been used to address the Cutting Stock Problem (CSP) which optimizes material usage in manufacturing [15, 16]. However, the interdisciplinary nature of these developments has resulted in a focus on enhancing explainability. For instance, augmented reality (AR) has been utilized to improve quality control procedures in panelized construction [7, 17]. In this approach, vision-based projection alignment methods enable CAD drawings to be superimposed onto construction panels. These techniques, employing image processing and spatial alignment algorithms, ensure precise alignment between virtual and physical elements, thereby increasing the accuracy of panel installations [7].

In addressing the optimization challenges associated with panelization, it is relevant to examine research on the 2DBP problem [9]. This is due to the fact that both scenarios, at a high-level, involve packing rectangular objects into a predefined space that minimizes wastage. In work similar to the proposed PAAD framework, an algorithmic solution to the 2DBP problem was developed in two phases: 1) identifying valid positions for each item, and 2) arranging these positions in a non-overlapping layout [18]. Initially, a pseudo-polynomial algorithm generate labels that represent the possible placements of each item within the bin. From this, integer linear programming applies various constraints in the second phase to ensure an optimal arrangement that prevents item overlap [18]. To improve the explainability of such approaches, human-derived heuristics (HDHs) have been integrated into evolutionary algorithms for 2DBP [19]. This utilizes heuristics generated by human annotators who had attempted the 2DBP problem, incorporating their choices as decision trees within the

evolutionary algorithm. As a result, traditional mutation and crossover operators were substituted with these heuristics to better replicate real-world practices.

These examples highlight a trend towards integrating computational methods with human insights in construction panelization and optimization. Specifically, PAAD will focus on four distinct considerations in a novel genetic evolutionary strategy based on the 2DBP, including: 1) generative design techniques for panelization, 2) the integration of manufacturing constraints within the design framework, 3) improved explainability via simple problem definitions and a preliminary interface, and 4) the ability to add post-processing rules that reflect revisions typically made by domain experts. Moreover, the approach is designed to manage complex wall structures featuring multiple openings and angled walls. This represents a distinction over traditional mathematical models and BIM-generative design frameworks, as PAAD offers greater flexibility in generating solutions while still adhering to the complexities of the problem space without needing explicit optimization definitions [12, 13]. To demonstrate the applicability of the proposed method in real-world settings and to provide future research with detailed information, the results are supplemented with visualizations in S1–S25 Figs.

## Methodology

### Datasets

**Automated baseline.**   In the current workflow of our industry partner, panelization is primarily managed through a rule-based approach, implemented as a plugin for Autodesk Revit [20]. This plugin, when applied to a CAD file, determines an optimal panelization solution. The results can either be exported or visualized, displaying the number of panels used. To provide a benchmark of the existing process, a dataset comprising of automated baselines was created, along with the actual configuration that was used in production. This dataset gives an insight into the initial solutions before any modifications by domain experts to show how the proposed work could improve over current methods. Due to the proprietary nature of this software, it is not open-source but can be described as a rule-based algorithm that ensures the solution will be structurally sound, regardless of its optimality in terms of material usage. As a result, there is an opportunity to outperform these outputs by considering both structural and material wastage aspects.

**Manual baseline.**   After receiving the initial panelization results, our partner's domain experts will typically refine the output to enhance optimality. This manual adjustment process is time consuming, as the most optimal configuration is often uncertain, especially when considering complexities such as angled walls or openings. If the automated method could consistently yield satisfactory solutions, it would reduce the need for these efforts and reduce the time spent planning for manufacturing. Therefore, a collection of manual baselines was established to assess the practical impact of the proposed work, by having three domain experts produce potential solutions. This also demonstrated that there can be several acceptable outputs for a given wall and that it is difficult to define optimality.

**Repository.**   To prepare the building plans used in our automated and manual baselines, the following details were manually encoded: 1) overall width, 2) overall height, 3) start and end coordinates of any angled lines, and 4) dimensions and coordinates of openings. To overcome the issues of data confidentiality in this domain, these details were made publically available along with a synthetic data generator and compiled version of PAAD on a Github repository (https://github.com/andrfish/PAAD). This generator enables the production of random walls to demonstrate the data format, and can also create irregular shaped walls to show

the versatility of the proposed work. It also demonstrates the simplistic formatting that is used as input to further enhance the algorithm's generalizability.

## 2D bin packing

To provide context for PAAD, it is important to understand the concept of 2DBP [9] and how it relates to panelization. The basic 2DBP problem involves two key components: 1) a bin, and 2) a set of items. The bin is defined as a rectangular container with a pre-defined width and height, designed to accommodate the set of items. Each item in the set is also pre-defined by its width and height, which is smaller than the bin's dimensions [9, 18]. Using these definitions, the typical goal is to fit a majority of these items into the bin in order to fill as much of the space as possible. For example, consider the fitness function shown in Eq 1 that such an algorithm may try to minimize:

$$
fitness = (bin.width \times bin.height) -
$$
$$
\sum_{i=0}^{len(items)} \begin{cases} (items[i].width \times items[i].height) & ; if \ items[i].packed \\ 0 & ; otherwise \end{cases}
\tag{1}
$$

where the *fitness* would represent how much area is unfilled by the *items* that have been *packed* within the *bin*.

When applying this concept to panelization, the *bin* represents a wall while the set of *items* symbolizes panels. Then, the following modifications are required that add complexity to the methodology: 1) allowing for a dynamic set of panels, 2) defining openings within the wall, and 3) accommodating angled walls. In this scenario, the set of panels are not pre-defined and the algorithm instead works to generate an optimal set that conforms to the manufacturing limitations. Conversely, openings such as windows or doors are treated as predefined items with static placement within the wall, forcing the algorithm to pack around them. Similarly, to manage angled walls, static objects can be utilized to fill void areas and guide the packing process. The placement of these objects is important, however, as it will effect how the panelization algorithm is able to fill the wall since each panel will still need to be rectangular. This is addressed by ensuring that each opening has a void area directly above it if applicable, and that the gaps between are filled with a static object that is a size less than or equal to the maximum manufacturing constraints. To visualize the changes that need to be made to the 2DBP, consider Fig 2 which shows how the goal of the algorithm is modified. First, stationary objects are introduced which are used to accommodate openings as well as angled lines. Then, a dynamic item set is introduced to accommodate manufacturing constraints and change the goal to generating the minimal number of items needed to pack the bin or wall.

Consider an example where a wall *W* is specified with the following dimensions: *W.width* = 150" and *W.height* = 100". Within wall *W*, two openings are defined in a set *O*: *O*[0].*x* = 25", *O*[0].*y* = 10", *O*[0].*width* = 15", *O*[0].*height* = 20" and *O*[1].*x* = 85", *O*[1].*y* = 20", *O*[1].*width* = 30", *O*[1].*height* = 15". The *x* and *y* coordinates determine the location of the bottom left corner of each opening within wall *W*. Additionally, an angled line *A* at the top of *W* starts at coordinates *A.start_x* = 0", *A.start_y* = 85" and ends at *A.end_x* = 150", *A.end_y* = 100". To accommodate the angle and facilitate 2DBP, void spaces are initially created above each opening. The surrounding space is subsequently filled to ensure complete coverage above line *A*, with these voids represented in the array *V*. This configuration is depicted in Fig 3, illustrating how the problem space of *W* has been adjusted for 2DBP by considering *O* and *V* as prepacked, stationary items.

**Goal:** Maximize the number of items placed into the bin

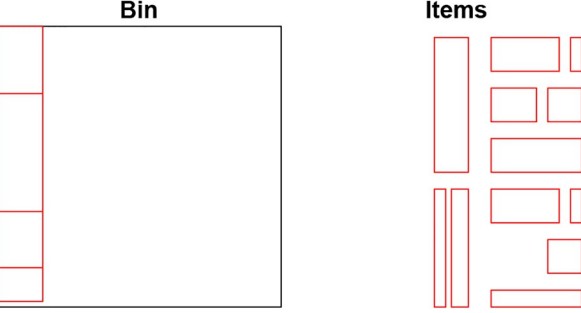

**(a)** A regular 2DBP where the goal is to maximize the number of items placed into a bin

**Goal:** Maximize the number of items placed into the bin that respect the stationary items

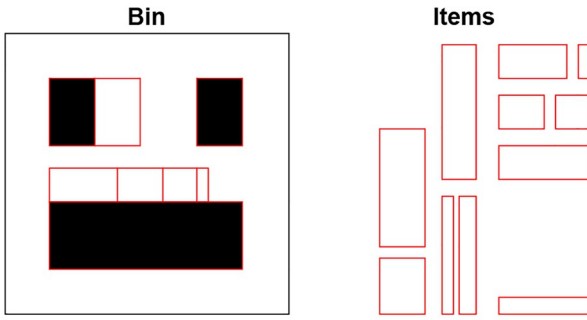

**(b)** A modified 2DBP with stationary objects to accommodate openings as well as angled lines

**Goal:** Generate the minimal number of items needed to fill all of the available space within the bin

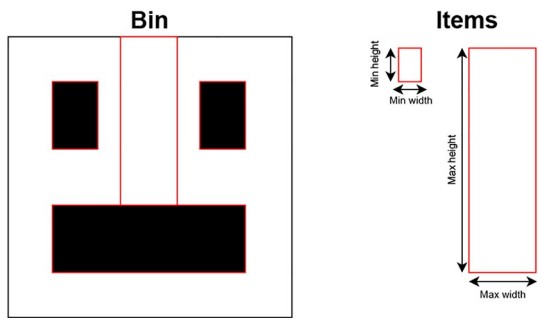

**(c)** A modified 2DBP for PAAD to accommodate manufacturing constraints and change the goal to generating a minimal number of items needed to pack the bin

**Fig 2. 2DBPP modifications.** An overview of the modifications that need to be made to the 2DBP for PAAD, beginning with the original problem definition in (a), a variation in (b) that accomodates openings as well as angled lines for the panelization process, and the final definition in (c) which accommodates manufacturing constraints as well as changes the optimization goal to a minimization problem.

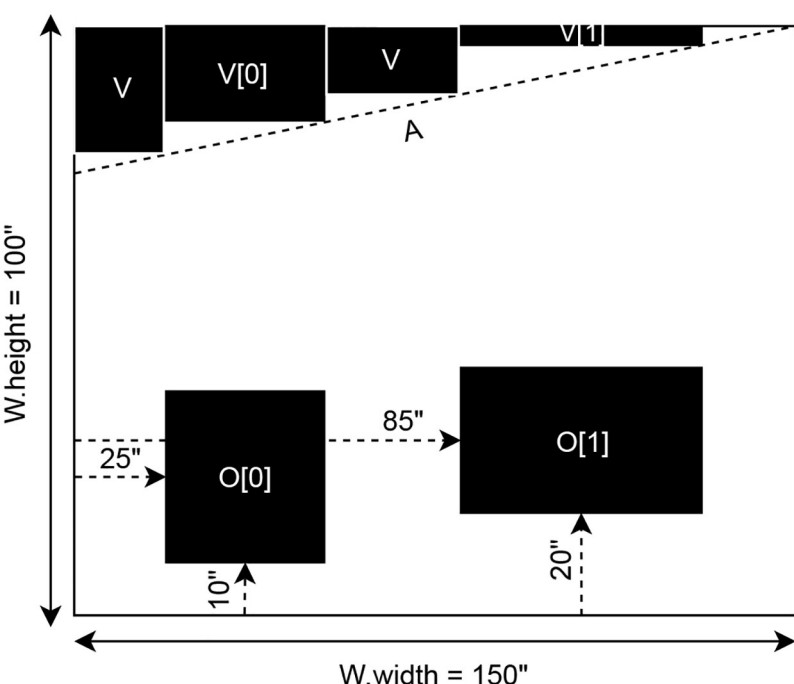

**Fig 3. Notation example.** An example wall using the notation for wall $W$, openings $O$, angled line $A$, and void space $V$ to visualize how the problem space for PAAD is defined.

### Genetic algorithm

From the adapted concept of 2DBP, it becomes apparent that there may be multiple viable solutions to panelize a given wall. To streamline this complexity and converge on an optimal configuration, PAAD incorporates a novel genetic evolutionary algorithm [8]. To the best of our knowledge, such an approach has not been applied to the panelization problem, nor are there publicly available baselines that it can be compared with. Through the description of this methodology, it will be observed that the use of an evolutionary algorithm has the advantage over pre-existing implementations by being able to evaluate several possible solutions at once, accommodate manufacturing constraints, allow for post-processing to incorporate structural constraints, and handle complex problem spaces. This is more complex than traditional evolutionary approaches as the panelization problem requires these considerations in order to produce a feasible solution for real-world use. These advantages have been observed in the domain for construction related tasks such as job shop scheduling, foundation design optimization, and structural engineering [21–23]. Furthermore, Algorithms 1–2 are also provided to understand how populations are initialized and processed in the evolutionary cycle, with explanations provided later on.

**Constraints.** To integrate the proposed genetic evolutionary algorithm [8] into our industry partner's existing workflow, establishing a framework of constraints was essential. This aimed to replace their extensive rule-based system which had been developed to ensure structural integrity and adherence to manufacturing limitations. To accommodate this, PAAD's main constraint was that it needed to use the following three panels types: 1) standard panels ($T_s$), 2) header panels ($T_h$), and 3) horizontal panels ($T_p$). Each panel type was subject to specific size constraints, defining their allowable dimensions. The standard panel ($T_s$) ranged from a minimum of 1 1/4"$W$ × 5"$H$ to a maximum of 48"$W$ × 143"$H$. Similarly, the header

panel ($T_h$) was limited to dimensions from 1 1/4"$W \times 5$"$H$ to 144"$W \times 47$"$H$, and the horizontal panel ($T_p$) from 1 1/4"$W \times 5$"$H$ to 144"$W \times 48$"$H$. These are visualized in Fig 4 to demonstrate their differences further.

An additional constraint was implemented to enhance the decision-making process concerning panel type selection, focusing on the panels' placement and height. The standard panel ($T_s$) was designated as the default choice under all circumstances. When a panel was required above an opening, a header panel ($T_h$) was chosen to ensure the necessary structural support. Conversely, when the needed panel height was below 48", a horizontal panel ($T_p$) was selected as the more appropriate option. This systematic method for choosing panel types was aimed at optimizing structural integrity, ensuring that each panel type was used in situations that matched its design and functional attributes.

However, there are several other rules considered in the current industry practices that PAAD does not explicitly incorporate. These mostly address outlier scenarios such as irregular geometries (which are addressed through the use of stationary objects and void spaces), but can also be critical for maintaining structural integrity in instances where a company is producing their own proprietary material. To accommodate this and ensure that the final result conforms with industry-specific rules, PAAD allows for the definition of customizable post-processing steps to refine the final output. For instance, our industry partner requires that header panels ($T_h$) must extend by 5" beyond either side of an opening, although it is permissible for a large panel to cover multiple openings. Instead of limiting the genetic algorithm with this requirement, the final output from PAAD can be algorithmically reviewed to confirm that each panel adheres to these criteria, through merging or adjusting the width as needed. This approach provides greater flexibility over the final product while still considering the complexities of the panelization problem, without the constraints associated with a rule-based system.

**Populations.** Due to the ambiguity of the problem definition, there exist multiple optimal or nearly optimal solutions for panelization. In such scenarios, a genetic algorithm is beneficial as it evaluates and evolves various solutions or populations simultaneously, adapting to identify the most efficient layout [11]. In the proposed PAAD framework, unlike in standard 2DBP problems [10], the items or panels are dynamically generated, conforming to predefined

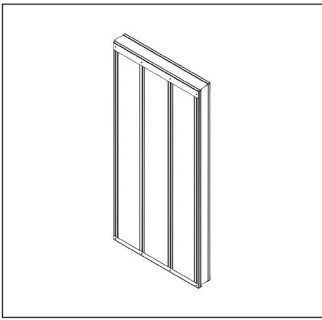 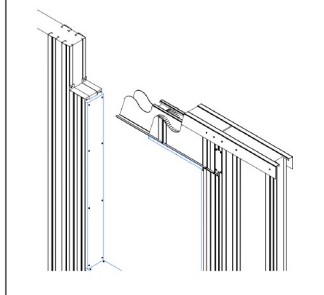 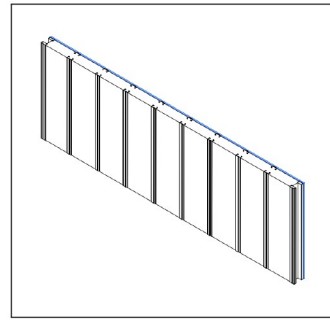

**(a)** A standard panel ($T_s$) which is subject to a minimum size of 1 1/4"$W \times 5$"$H$ to a maximum of 48"$W \times 143$"$H$

**(b)** A header panel ($T_h$) which is subject to a minimum size of 1 1/4"$W \times 5$"$H$ to a maximum of 144"$W \times 47$"$H$

**(c)** A horizontal panel ($T_p$) which is subject to a minimum size of 1 1/4"$W \times 5$"$H$ to a maximum of 144"$W \times 48$"$H$

**Fig 4. Panel types.** A visualization of the different panel types considered by PAAD, with (a) showing a standard panel, (b) a header panel, and (c) a horizontal panel.

minimum and maximum manufacturing constraints. Additionally, the optimization goal diverges from the traditional objective of maximizing item quantity, aiming instead to generate the minimal number of panels required to completely fill a given wall. This complexity is increased further by the necessity to integrate architectural elements such windows and doors (i.e., openings) into the panels' placement and dimensions. As a result, the proposed work is more complex than traditional evolutionary approaches due to these requirements that were necessitated by industry standards.

To initialize the populations, PAAD generates sets of potential panelization solutions as described in Algorithm 1. Through a loop, panels of random sizes within the manufacturing constraints are created (line 9), and then assessed for their ability to fit within the specified wall (line 13). The loop concludes once the total area of the fitted panels equals or surpasses the wall's area. Subsequently, all panels within the population, including those not initially placed, are organized by size, with preference given to larger panels (line 20). This prioritization supports PAAD's objective of reducing the total number of panels by initially considering the largest ones, which cover more area.

Given that PAAD's initialization method is random (with adherence to manufacturing constraints) and designed to consider features such as openings and angles, it was important to ensure that simpler problem spaces could be resolved quickly. For instance, a rectangular wall without openings can be systematically optimized by adhering to manufacturing constraints to occupy the space with the largest feasible panels. Therefore, an additional population that solely accounts for the overall dimensions, disregarding any openings or angled segments, is created to derive an optimal solution. This approach may also expedite convergence on marginally more complex configurations, such as those featuring a single opening, since the solution would already be partially completed [9, 13]. For example, assume there is a wall with a *width* = 100" and *height* = 100", that contains 1 opening with a *width* = 10" and *height* = 10". Since the opening covers 10% or (10" × 10")/(100" × 100") of the wall, a solution that ignores all complexities has the possibility of solving up to 90% of the problem space.

**Algorithm 1** Evolutionary Population Initialization

```
Input: wall = an initialized wall that contains information about the
wall.area, openings, as well as angled lines, min_panel_dimensions =
an array with the minimum width and height that a panel can be, and
max_panel_dimensions = an array with the maximum width and height that
a panel can be
Output: population = an array of randomly generated panels with a sub-
set that has been verified to fit within the wall
 1: procedure init_population(wall, min_panel_dimensions,
               max_panel_dimensions)
 2:
 3:   # Initialize an empty population
 4:   population ← empty_array()
 5:   total_area ← 0
 6:
 7:   # Create panels until the wall has been filled
 8:   while total_area < wall.area do
 9:     panel ← new_random_panel(min_panel_dimensions,
       max_panel_dimensions)
10:     population.append(panel)
11:
12:     # Check that the current panel would fit within the wall given
     the packed panels in the population
13:     if fits_in_wall(wall, panel, population) then
14:       panel.packed ← true
```

```
15:        total_area+ = panel.width × panel.height
16:      end if
17:   end while
18:
19:   # Set the packed property of each panel to false, order the array
       from largest to smallest, and return the population
20:   unpack_and_sort_panels(population)
21:   return population
22: end procedure
```

## Evaluation

With the constraints and population procedures established, the genetic evolutionary algorithm initiates by evaluating variables that describe the problem space. These variables include the wall's width, height, openings and angled lines, which are important for comprehending the wall's geometry relevant to panelization. Openings are characterized by arrays detailing their width, height, and *x* and *y* coordinates, similar to angled lines that provide information on their starting and ending coordinates. This was preferred over BIM definitions as it is more simplistic and can allow for projects at different stages in the design process to be panelized. Using these definitions alongside a set of initialized populations, the next step involves evolving the current solutions followed by an assessment to identify the most optimal set.

To evolve the set of panelization solutions (i.e., populations), there are two possibilites that can occur, which is described in Algorithm 2: 1) mutation and 2) crossover. In the case of mutation, each panel faces a probability assessment to determine if its width or height will be altered randomly (line 9). This adjustment is inclined towards increasing the dimensions, aligning with the goal of minimizing the total number of panels required. In the case of crossover, each population undergoes a probability assessment to determine whether it will combine with another population in the current set (line 20). This process allows for the potential introduction of more optimal panels between solutions, thereby enhancing the overall result [24, 25].

During the evaluation phase, each panel within a given population is assessed from largest to smallest to determine if adequate space exists for its placement in the wall. This involves searching for an unoccupied area starting from the top left corner, while ensuring no overlap with already placed panels, stationary objects (i.e., openings), or voids above angled lines. Upon identifying a suitable location, the panel is added to the solution, and the loop proceeds to the next item. This process is repeated for the entire population, and upon completion, any remaining space on the wall that meets the minimum manufacturing constraints is filled with a correspondingly sized panel. This ensures the solution's feasibility for real-world application, as all spaces must be utilized to maintain structural integrity.

**Fitness.** Following the evaluation stage, panels that were not successfully placed undergo further random mutations [26]. For panels that have been positioned, the algorithm attempts to enlarge them as much as possible without causing overlaps. It also explores merging opportunities, enabling adjacent panels to join or partially merge, which helps decrease the total number of packed items. This procedure adheres to manufacturing constraints, ensuring that the final dimensions do not exceed maximum limits and that the resulting shape remains rectangular (i.e., the adjacent panels must have equal heights). The fitness of each population is then determined using a formula that calculates the area covered by the placed panels, adjusted by the number of placed panels [27]. The goal is to minimize this fitness value, aiming for a

solution that covers the largest area with the fewest panels, as described in Eq 2:

$$
\begin{aligned}
fitness = {}& (wall.width \times wall.height) - \\
& \sum_{i}^{len(openings)} (openings[i].width \times openings[i].height) - \\
& \sum_{j}^{len(voids)} (voids[j].width \times voids[j].height) - \\
& \sum_{k=0}^{len(panels)} \begin{cases} (panels[k].width \times panels[k].height) & if\ panels[k].packed \\ 0 & otherwise \end{cases} \\
& + len(\{panel \mid panel \in panels,\ panel.packed == true\})
\end{aligned}
\tag{2}
$$

where the wall's area is calculated (*wall.width × wall.height*), the area of the openings is deducted since they cannot be panelized ($\sum_{i}^{len(openings)}(openings[i].width \times openings[i].height)$), the area of the voids is deducted ($\sum_{j}^{len(voids)}(voids[j].width \times voids[j].height)$), the area covered by panels is subtracted to determine wall coverage ((*panels[k].width × panels[k].height*) *if panels[k].packed*), and the total count of packed panels is added to favor solutions that use fewer panels (*len({panel | panel ∈ panels, panel.packed == true})*).

**Algorithm 2** Evolutionary Cycle

```
Input: Populations = an array of population objects that is in the
current evolution, initialized as per Algorithm 1, mutation_prob = a
probability that a panel will be mutated, and crossover_prob = a prob-
ability that a population will crossover with another set of panels
Output: populations = an updated version of the populations that has
subjected each population to mutation_prob and crossover_prob
probabilities
 1: procedure evolve_panels(populations, mutation_prob,
            crossover_prob)
 2:   # Cycle through each population in the populations
 3:   for each i ∈ len(Populations) do
 4:     population ← Populations[i]
 5:     # Cycle through each panel in the population
 6:     for each j ∈ len(population) do
 7:       panel ← population[j]
 8:       # Randomly mutate either the width or height
 9:       if random_float() < mutation_prob then
10:         if random_float() ≤ 0.5 then
11:           panel ← mutate_width(panel)
12:         else
13:           panel ← mutate_height(panel)
14:         end if
15:       end if
16:     end for
17:
18:     # Randomly crossover with another population
19:     if random_float() < crossover_prob then
20:       population ← crossover_random_population(population,
            populations)
21:     end if
22:   end for
23:   return populations
24: end procedure
```

If no significant improvement in fitness is observed across several generations, PAAD implements additional measures to ensure it is not stuck in a local minimum [28, 29]. This involves integrating a modified version of the fittest population back into the solution set multiple times, which includes only the successfully placed panels (i.e., any unplaced panels from the evaluation phase are removed). To these new populations, several smaller-sized panels are randomly added, ensuring their combined potential coverage does not substantially exceed the wall's total area. These smaller panels are designed to fill additional space, gradually expanding through mutations once placed by continuously increasing their width and height until an overlap is detected or the bounds of the wall are exceeded. Should the fitness not enhance after these adjustments, the procedure is reiterated with an additional measure: all placed panels are subjected to a random mutation, biased towards slightly decreasing their dimensions.

When it becomes apparent that the genetic algorithm cannot further optimize the problem space, or when a predefined stopping condition, such as a maximum number of evolutions, is met, the population with the highest fitness is selected as the final solution. However, due to the complexity of identifying the most optimal configuration, this method has the inherent limitation of possibly not achieving the best solution overall. This limitation can be partially mitigated by consulting industry experts to review the final output, which is a necessary quality control step for real-world applications before proceeding to manufacturing.

To summarize the optimization goal at a high-level, consider the different panel types as indices where $0 = T_s$, $1 = T_h$, and $2 = T_p$. Next, associated constraints include a minimum width constraint of $min\_width = 11/4"$, a minimum height constraint of $min\_height = 5"$, a set of maximum width constraints ($max\_width_0 = 48"$, $max\_width_1 = 144"$, $max\_width_2 = 144"$), and a set of maximum height constraints ($max\_height_0 = 143"$, $max\_height_1 = 47"$, and $max\_height_2 = 48"$). These constraints guide the configuration of the panels as detailed in Algorithms 3–4, and are defined by the optimization function in Eq 3:

$$\underset{P,W,O,V,M,C}{\text{minimize}} \quad fitness(P, W, V, O)$$

$$\text{subject to}$$

$$P = \{p \mid p \in P, evolve\_panels(p, M, C)\}, \quad Population\ evolution, \tag{3}$$

$$c_{width}(P) = true, \qquad\qquad\qquad Panel\ width\ constraint,$$

$$c_{height}(P) = true, \qquad\qquad\qquad Panel\ height\ constraint$$

where $P$ is a set of population objects $p$ that were each initialized using Algorithm 1, $W$ is the wall definition (i.e., height and width), $O$ is the openings definition (i.e., where each opening has a height, width, starting x-coordinate, and starting y-coordinate), $V$ is the void area definitions (i.e., for space above angled lines), $M$ is a mutation probability, $C$ is a crossover probability, $fitness$ represents a function that evaluates Eq 2 for each $p$ in $P$ to determine the minimum result, and $evolve\_panels$ represents Algorithm 2. To summarize their definitions from before, Algorithm 1 initializes populations by generating potential panelization solutions within manufacturing constraints, Eq 2 assesses each population's fitness with the aim of covering a wall using the fewest panels, and Algorithm 2 evolves these populations through mutations that alter panel dimensions and crossovers that merge potential solutions.

```
Algorithm 3 Panel width constraint
procedure c_width(P)
  for each population ∈ P do
    for each panel ∈ population do
      if panel.width > max_width_panel.type ||
        panel.width < min_width then
```

```
        return false
      end if
    end for
  end for
  return true
end procedure
Algorithm 4 Panel height constraint
procedure c_height(P)
  for each population ∈ P do
    for each panel ∈ population do
      if panel.height > max_height_panel.type ||
        panel.height < min_height then
          return false
      end if
    end for
  end for
  return true
end procedure
```

## Assumptions

The methodology of PAAD operates under several key assumptions about the problem space: 1) the dimensions of the wall are known, 2) the minimum and maximum sizes of the panels are known, 3) the presence of any angled lines in the wall are known, 4) panels cannot overlap each other, and 5) openings or void spaces must not be covered by panels. We consider these assumptions to not be restrictive for our proposed approach, as they mirror the conditions and practices typically encountered in real-world scenarios. To validate the effectiveness of PAAD, the next section presents two baselines that compare the algorithm's performance against current industry standards as well as domain experts.

## Results

### Automated baseline

To evaluate PAAD's effectiveness, our industry partner provided 24 architectural drawings that were previously panelized using their proprietary software or rule-based system, alongside the final solutions used in production. Initially, the problem definitions for these walls were manually encoded to specify: 1) the dimensions, 2) any openings, and 3) any angled lines. Subsequently, PAAD was executed until it reached a minima and could no longer enhance the fitness. The results, presented in Table 1, indicate that the proposed method either matched or exceeded the performance of the current software in 22 out of 24 scenarios, with 10 of these 22 instances also surpassing human annotations. This comparison was based on the number of panels used in the final solutions, which were verified for feasibility by an industry expert before any post-processing steps. These findings suggest that genetic evolutionary algorithms can yield optimal outcomes for panelization with fewer predefined rules. For added clarity, visual representations of the scenarios are provided in S1–S22 Figs, with two of them highlighted in Figs 5 and 6. Here, it can be observed how PAAD is able to generalize across designs of varying complexities, whereas the rule-based system faces difficulties. For example, consider Fig 5 where the problem space is a simple rectangular wall with three rectangular openings. Conversely, in Fig 6, there is an angled wall along with several angled openings that can be difficult for a rule-based approach to optimally solve. Referring to Table 1, PAAD was able to outperform the rule-based system in both of these scenarios (i.e., 10 and 20, respectively), with a significant difference in number of panels used for the latter scenario.

**Table 1. The number of panels used in the automated baseline.**

| Scenario | Rule-Based System | Domain Expert | PAAD |
|---|---|---|---|
| 1 | 11 | 7 | 8 |
| 2 | 66 | 59 | **59** |
| 3 | 12 | 12 | **12** |
| 4 | 9 | 11 | **9** |
| 5 | 25 | 21 | **21** |
| 6 | 17 | 17 | **17** |
| 7 | 34 | 24 | **23** |
| 8 | 75 | 83 | **71** |
| 9 | 17 | 17 | **17** |
| 10 | 15 | 15 | **14** |
| 11 | 31 | 19 | **19** |
| 12 | 16 | 16 | **12** |
| 13 | 12 | 15 | 16 |
| 14 | 12 | 12 | **10** |
| 15 | 10 | 10 | **10** |
| 16 | 29 | 22 | **22** |
| 17 | 21 | 21 | **20** |
| 18 | 19 | 20 | **18** |
| 19 | 32 | 32 | 33 |
| 20 | 45 | 25 | **22** |
| 21 | 18 | 18 | **18** |
| 22 | 18 | 18 | **18** |
| 23 | 13 | 13 | **13** |
| 24 | 14 | 15 | **13** |

## Manual baseline

To illustrate that panelization problems may yield multiple optimal solutions, additional five projects were assessed by three domain experts, alongside the rule-based method. These assessments were conducted independently to ensure unbiased recommendations, aiming for

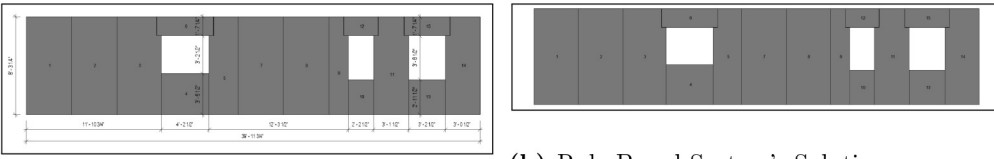

**(a)** Domain Expert's Solution

**(b)** Rule-Based System's Solution

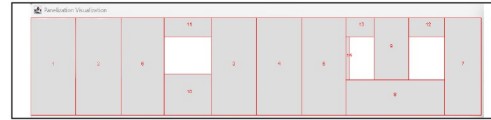

**(c)** PAAD's Solution

**Fig 5. Automated scenario 10.** The results of automated scenario 10, where (a) presents the domain expert's solution, (b) shows the rule-based system's output, and (c) shows PAAD's solution.

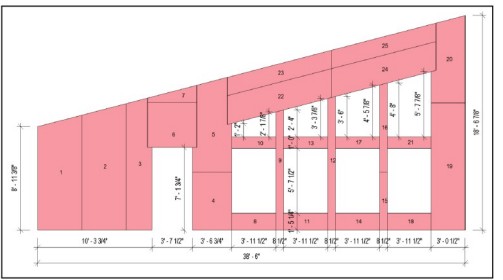

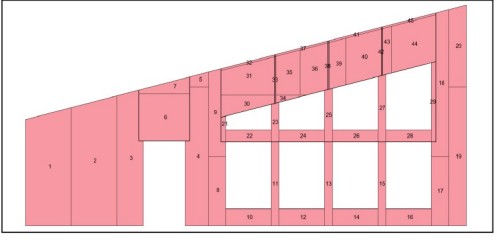

**(b)** Rule-Based System's Solution

**(a)** Domain Expert's Solution

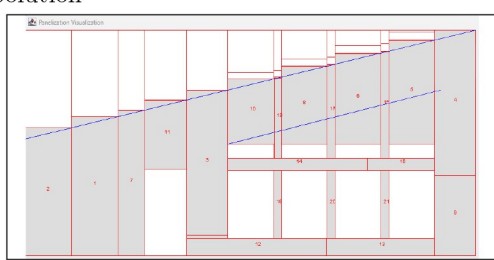

**(c)** PAAD's Solution

**Fig 6. Automated scenario 20.** The results of automated scenario 20, where (a) presents the domain expert's solution, (b) shows the rule-based system's output, and (c) shows PAAD's solution.

minimal panel usage similar to the evolutionary strategy. The findings, detailed in Table 2, reveal that the experts generally used an identical number of panels but differed in their placement strategies within the walls. Moreover, the evolutionary algorithm consistently matched or outperformed the manual assessments and exceeded the software's results in 4 out of 5 cases. This indicates that PAAD can reliably generate optimal solutions across various problem sets, while the conventional software may struggle due to adherence to a complex rule set. The outcomes of the first two scenarios are highlighted in Figs 7 and 8, with the remaining results available in S23–S25 Figs. Similar to the observation made in the automated baseline, PAAD is able to consistently produce optimal solutions across problem spaces of varying complexities. In the highlighted scenarios (i.e., Figs 7 and 8), the rule-based system produced solutions with 31 and 17 panels, respectively, whereas the proposed work converged on 18 and 14 panels, demonstrating its robustness.

## Preliminary interface

To demonstrate how PAAD could be integrated into a company's existing workflow, a preliminary interface has been developed. This interface initially applies line segment detection as

**Table 2. The number of panels used in the manual baseline.**

| Scenario | Domain Expert 1 | Domain Expert 2 | Domain Expert 3 | Rule-Based System | PAAD |
|---|---|---|---|---|---|
| 1 | 22 | 21 | 21 | 31 | 18 |
| 2 | 13 | 13 | 14 | 17 | 14 |
| 3 | 16 | 16 | 16 | 20 | 15 |
| 4 | 13 | 14 | 14 | 17 | 13 |
| 5 | 5 | 5 | 5 | 5 | 5 |

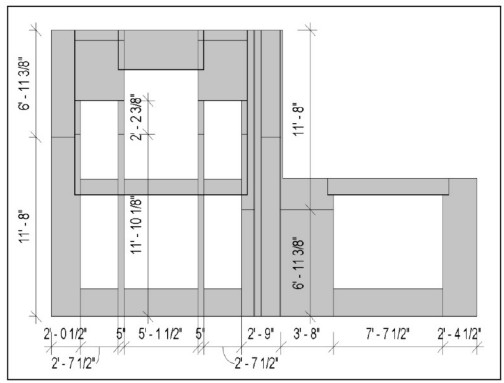

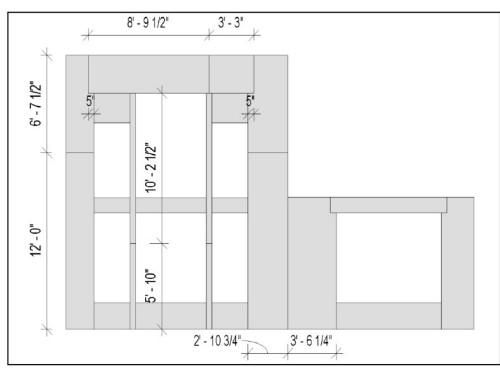

**(a)** Rule-Based System's Solution

**(b)** Domain Expert 1's Solution

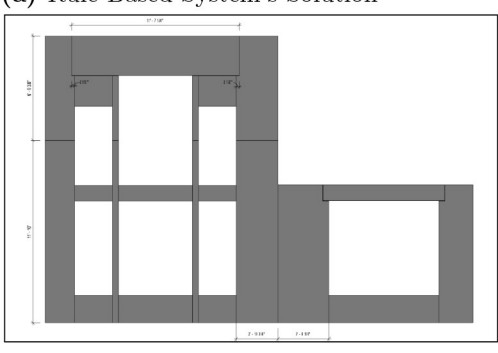

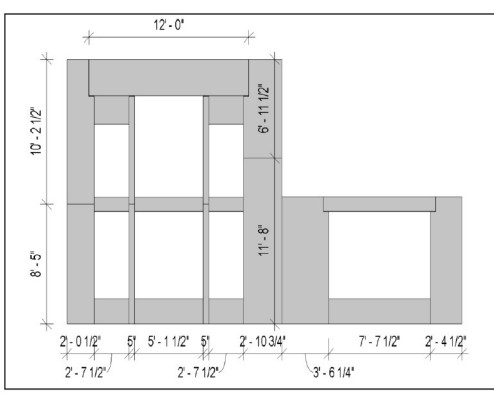

**(c)** Domain Expert 2's Solution

**(d)** Domain Expert 3's Solution

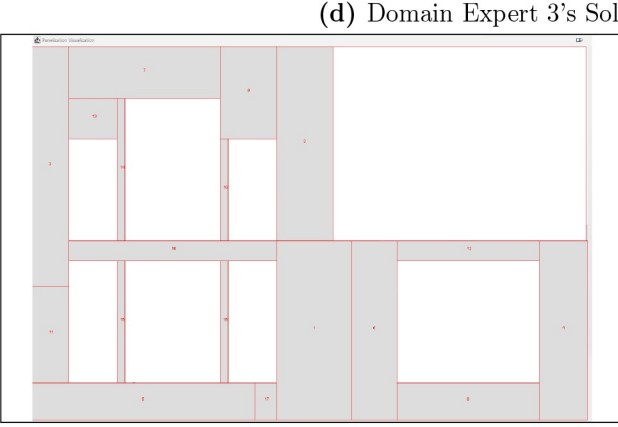

**(e)** PAAD's Solution

**Fig 7. Scenario 1.** The results of manual scenario 1, where (a) shows the rule-based system's output, (b)—(d) presents three domain experts' solutions, and (e) shows PAAD's solution.

well as text recognition techniques [30, 31] to determine the problem definitions based on an architectural drawing, then initiates the evolutionary process. As the algorithm evaluates results in real-time, the interface continuously refreshes to display the fittest solution from each evolution. These solutions are superimposed on the architectural drawing to improve

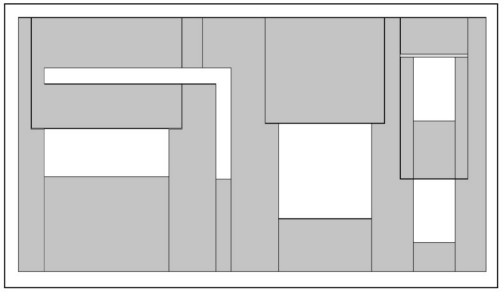

**(a)** Rule-Based System's Solution

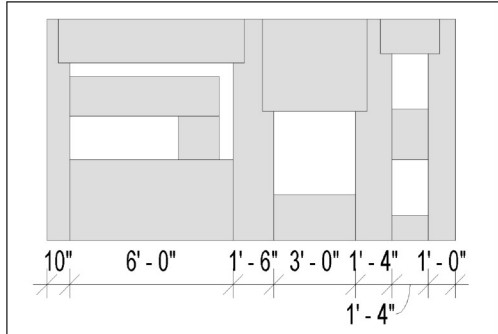

**(b)** Domain Expert 1's Solution

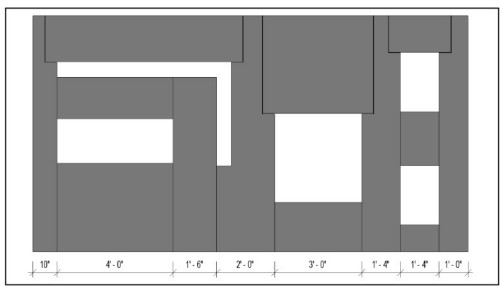

**(c)** Domain Expert 2's Solution

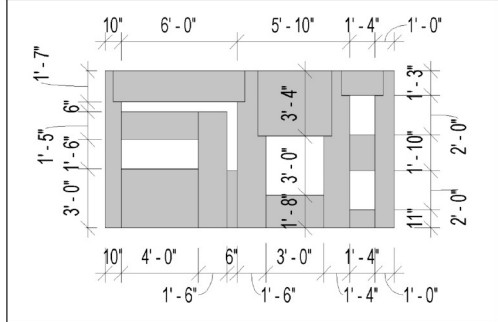

**(d)** Domain Expert 3's Solution

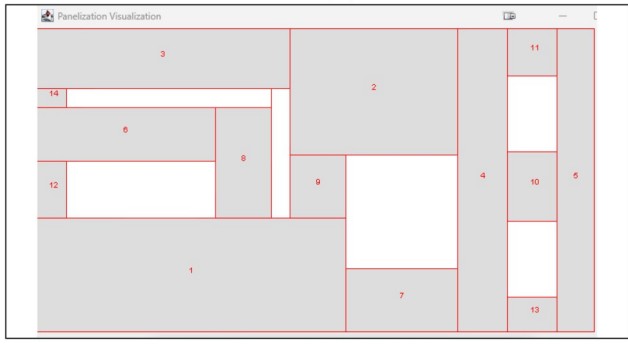

**(e)** PAAD's Solution

**Fig 8. Scenario 2.** The results of manual scenario 2, where (a) shows the rule-based system's output, (b)—(d) presents three domain experts' solutions, and (e) shows PAAD's solution.

explainability and provide detailed dimensional information about each panel, as illustrated in Fig 9. It can be observed that further refinements are needed to ensure that the overlay is accurate. However, users have the option to modify the results if necessary and export them for additional analysis.

## Discussion

The lack of publicly accessible datasets necessitated collaboration with an industry partner for PAAD's development, ensuring the algorithm's applicability in real-world scenarios by

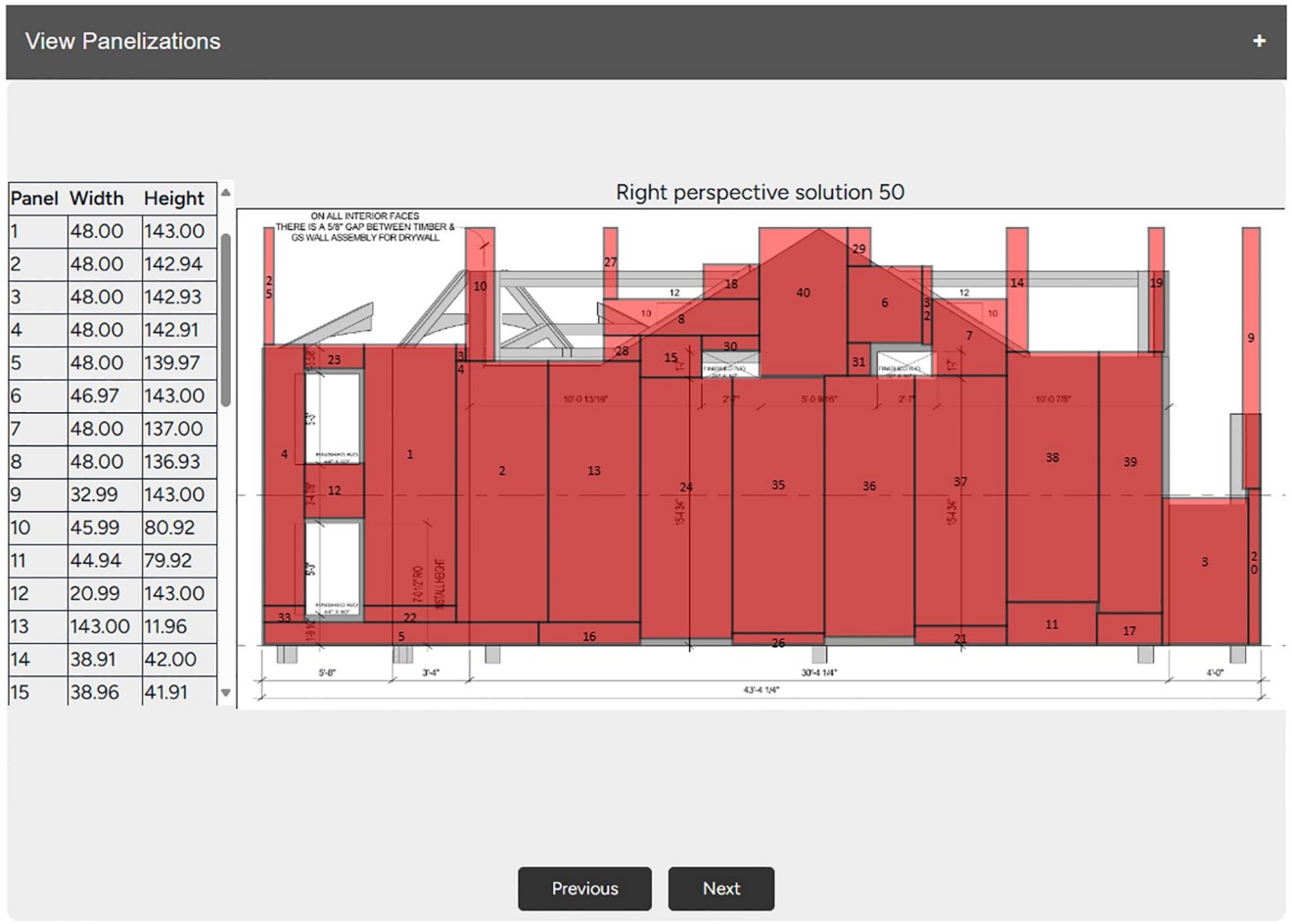

**Fig 9. An example from the preliminary interface where PAAD's results are overlaid onto an architectural drawing, with some noisy results.**

facilitating comparisons with both domain experts and proprietary software. Acknowledging the algorithms' potential to minimize waste and their environmental benefits, a synthetic data generator was created and the architectural drawing encodings were made public to reduce these restrictions, supplemented by a detailed Appendix. This contribution aims to simplify the formatting of input data, reducing the need for detailed definitions typically associated with BIM.

To increase the impact of panelization algorithms, future studies should investigate automated techniques for extracting architectural information. In both baseline comparisons, the dimensions, openings, and angled line details for each wall were manually encoded, allowing for direct benchmarking against domain experts. This intensive process required the examination of building plans and the definition of various dimensions. Therefore, an automated strategy could leverage line segment detection and text recognition techniques [30, 31] to streamline this information extraction, while still allowing for modifications before panelization to ensure accuracy. Although our initial attempts in this direction are showcased through the preliminary interface, it is evident that additional refinements are necessary to guarantee precise panel placement.

## Conclusion

In the construction industry, domain experts typically utilize rule-based methods to develop optimal plans before manufacturing a product. This approach is used in panelization, for example, which is the process of covering a building's walls with panels to form the overall structure. However, the presence of complexities, such as angled walls or openings, makes it challenging to achieve optimality and establish rules for such scenarios. Consequently, experts often need to manually review the results, which is a time-intensive task. To address these challenges, a genetic evolutionary approach called PAAD was developed in collaboration with an industry partner. PAAD is designed to comply with manufacturing constraints, modifies the traditional 2D bin packing problem to accommodate a dynamic array of items, and accommodates complexities within the problem space. It applies a genetic evolutionary strategy to concurrently explore multiple solution variations.

To assess its effectiveness, a dataset from our industry partner was utilized. This resulted in an automated baseline that was conducted on 24 architectural drawings to compare with the current industry software implementation and those used in production. The results revealed that PAAD matched or outperformed the software in 22 out of 24 scenarios, and improved upon the production configurations in 10 of these cases. Additionally, to understand the intricacies of panelization further, a manual baseline was established where three industry experts, alongside the software, assessed 5 architectural drawings. In this setup, PAAD proved to be competitive with the expert results and surpassed the software outcomes in 4 out of 5 cases. These findings demonstrate the potential of evolutionary approaches to replace traditional rule-based methods without sacrificing optimality. Furthermore, to provide motivation for future research, data repository and a preliminary interface were introduced. Our objective was to demonstrate the feasibility of integrating PAAD into real-world workflows and to highlight the advantages it offers to both the industry as well as its clients.

## Supporting information

**S1 Fig. Automated scenario 1.** The results of automated scenario 1, where (a) presents the domain expert's solution, (b) shows the rule-based system's output, and (c) shows PAAD's solution.
(TIF)

**S2 Fig. Automated scenario 2.** The results of automated scenario 2, where (a) presents the domain expert's solution, (b) shows the rule-based system's output, and (c) shows PAAD's solution.
(TIF)

**S3 Fig. Automated scenario 3.** The results of automated scenario 3, where (a) presents the domain expert's solution, (b) shows the rule-based system's output, and (c) shows PAAD's solution.
(TIF)

**S4 Fig. Automated scenario 4.** The results of automated scenario 4, where (a) presents the domain expert's solution, (b) shows the rule-based system's output, and (c) shows PAAD's solution.
(TIF)

**S5 Fig. Automated scenario 5.** The results of automated scenario 5, where (a) presents the domain expert's solution, (b) shows the rule-based system's output, and (c) shows PAAD's

solution.
(TIF)

**S6 Fig. Automated scenario 6.** The results of automated scenario 6, where (a) presents the domain expert's solution, (b) shows the rule-based system's output, and (c) shows PAAD's solution.
(TIF)

**S7 Fig. Automated scenario 7.** The results of automated scenario 7, where (a) presents the domain expert's solution, (b) shows the rule-based system's output, and (c) shows PAAD's solution.
(TIF)

**S8 Fig. Automated scenario 8.** The results of automated scenario 8, where (a) presents the domain expert's solution, (b) shows the rule-based system's output, and (c) shows PAAD's solution.
(TIF)

**S9 Fig. Automated scenario 9.** The results of automated scenario 9, where (a) presents the domain expert's solution, (b) shows the rule-based system's output, and (c) shows PAAD's solution.
(TIF)

**S10 Fig. Automated scenario 11.** The results of automated scenario 11, where (a) presents the domain expert's solution, (b) shows the rule-based system's output, and (c) shows PAAD's solution.
(TIF)

**S11 Fig. Automated scenario 12.** The results of automated scenario 12, where (a) presents the domain expert's solution, (b) shows the rule-based system's output, and (c) shows PAAD's solution.
(TIF)

**S12 Fig. Automated scenario 13.** The results of automated scenario 13, where (a) presents the domain expert's solution, (b) shows the rule-based system's output, and (c) shows PAAD's solution.
(TIF)

**S13 Fig. Automated scenario 14.** The results of automated scenario 14, where (a) presents the domain expert's solution, (b) shows the rule-based system's output, and (c) shows PAAD's solution.
(TIF)

**S14 Fig. Automated scenario 15.** The results of automated scenario 15, where (a) presents the domain expert's solution, (b) shows the rule-based system's output, and (c) shows PAAD's solution.
(TIF)

**S15 Fig. Automated scenario 16.** The results of automated scenario 16, where (a) presents the domain expert's solution, (b) shows the rule-based system's output, and (c) shows PAAD's solution.
(TIF)

**S16 Fig. Automated scenario 17.** The results of automated scenario 17, where (a) presents the domain expert's solution, (b) shows the rule-based system's output, and (c) shows PAAD's solution.
(TIF)

**S17 Fig. Automated scenario 18.** The results of automated scenario 18, where (a) presents the domain expert's solution, (b) shows the rule-based system's output, and (c) shows PAAD's solution.
(TIF)

**S18 Fig. Automated scenario 19.** The results of automated scenario 19, where (a) presents the domain expert's solution, (b) shows the rule-based system's output, and (c) shows PAAD's solution.
(TIF)

**S19 Fig. Automated scenario 21.** The results of automated scenario 21, where (a) presents the domain expert's solution, (b) shows the rule-based system's output, and (c) shows PAAD's solution.
(TIF)

**S20 Fig. Automated scenario 22.** The results of automated scenario 22, where (a) presents the domain expert's solution, (b) shows the rule-based system's output, and (c) shows PAAD's solution.
(TIF)

**S21 Fig. Automated scenario 23.** The results of automated scenario 23, where (a) presents the domain expert's solution, (b) shows the rule-based system's output, and (c) shows PAAD's solution.
(TIF)

**S22 Fig. Automated scenario 24.** The results of automated scenario 24, where (a) presents the domain expert's solution, (b) shows the rule-based system's output, and (c) shows PAAD's solution.
(TIF)

**S23 Fig. Manual scenario 3.** The results of manual scenario 3, where (a) shows the rule-based system's output, (b)—(d) presents three domain experts' solutions, and (e) shows PAAD's solution.
(TIF)

**S24 Fig. Manual scenario 4.** The results of manual scenario 4, where (a) shows the rule-based system's output, (b)—(d) presents three domain experts' solutions, and (e) shows PAAD's solution.
(TIF)

**S25 Fig. Manual scenario 5.** The results of manual scenario 5, where (a) shows the rule-based system's output, (b)—(d) presents three domain experts' solutions, and (e) shows PAAD's solution.
(TIF)

## Acknowledgments

The language in this manuscript was proof-read by a fine-tuned GPT-4 model from OpenAI [32]. It specifically looked at correcting only grammatical issues and did not provide any

theoretical contributions nor modify the text in any other way. Additionally, this project was done in collaboration with Greenstone Building Products in Brandon, Manitoba, Canada.

## Author Contributions

**Data curation:** Andrew Fisher.

**Formal analysis:** Andrew Fisher.

**Funding acquisition:** Muntasir Billah, Pawan Lingras, Jimmy Huang, Vijay Mago.

**Investigation:** Xing Tan.

**Methodology:** Andrew Fisher, Vijay Mago.

**Project administration:** Andrew Fisher.

**Resources:** Andrew Fisher.

**Software:** Andrew Fisher.

**Supervision:** Vijay Mago.

**Validation:** Andrew Fisher.

**Visualization:** Andrew Fisher.

**Writing – original draft:** Andrew Fisher.

**Writing – review & editing:** Andrew Fisher.

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
