## [Decision Letter · Decision Letter 0]

15 Apr 2024

PONE-D-24-12621PAAD: Panelization algorithm for architectural designsPLOS ONE

Dear Dr. Mago,

Thank you for submitting your manuscript to PLOS ONE. After careful consideration, we feel that it has merit but does not fully meet PLOS ONE’s publication criteria as it currently stands. Therefore, we invite you to submit a revised version of the manuscript that addresses the points raised during the review process.

We look forward to receiving your revised manuscript.

Kind regards,

Chen Li, Ph.D.

Academic Editor

PLOS ONE

“MITACS - IT22468”

Reviewers' comments:

Reviewer's Responses to Questions

**Comments to the Author**

1. Is the manuscript technically sound, and do the data support the conclusions?

Reviewer #1: Yes

Reviewer #2: Yes

Reviewer #3: Yes

Reviewer #4: Yes

2. Has the statistical analysis been performed appropriately and rigorously? 

Reviewer #1: Yes

Reviewer #2: N/A

Reviewer #3: N/A

Reviewer #4: N/A

3. Have the authors made all data underlying the findings in their manuscript fully available?

Reviewer #1: Yes

Reviewer #2: Yes

Reviewer #3: Yes

Reviewer #4: Yes

4. Is the manuscript presented in an intelligible fashion and written in standard English?

Reviewer #1: Yes

Reviewer #2: Yes

Reviewer #3: Yes

Reviewer #4: Yes

5. Review Comments to the Author

Reviewer #1: This research proposes the PAAD algorithm for optimizing panel configurations in architectural designs, addressing inefficiencies in rule-based methods. PAAD utilizes a genetic evolutionary strategy to minimize material waste while complying with manufacturing constraints. Validation against industry implementations and expert solutions demonstrates PAAD's effectiveness and potential to enhance efficiency in the construction industry. The topic of this research is significant.

What are the assumptions for using PAAD algorithm?

Reviewer #2: The work by Fisher and colleagues presented a novel algorithm called Panelization Algorithm for Architectural Designs (PAAD) to automate the optimal panel configuration. Their approach is built on the 2D bin packing method but engages a genetic evolutionary algorithm. The novelty of this method lies in the capability of adapting to both manufacturing rules (as hard constraints) and structural rules as well as post-processing rules (as flexible constraints) in a flexible way. Comparing against rule-based and domain expert solutions, they showed an overall advantage in PAAD over other methods.

Major issues:

1. Table 1 and 2, the performance of PAAD is compared with rule-based algorithms or domain expert in automated (Table 1) or manual (Table 2) tasks. It is definitely pivotal benchmark to prove the edge of PAAD, however, it might be also interesting to make some visual comparison between the methods in question. I saw such comparisons in Supporting Information, and maybe some can be moved to the main figures to enhance the benchmark.

2. Figures need better captions and description. The figure captions in current manuscript are overly succinct. For example, Fig 4/5, “Scenario 1/2. The results of manual scenario 1/2” but each figure contains five panels (a-e). Detailed description of each subfigure is needed here for the ease of readership.

3. The authors made public the source code at GitHub, which is great. However, I found two issues with the current version. First, the graphical interface has no documentation—only a license can be found at https://github.com/andrfish/PAAD at the moment. Second, when I ran the Java program (via the command line java Generator.java), I could only see the wall with random voids above the angled line and stationary openings (please see Review Fig 1-2). However, I could not obtain solutions “superimposed on the architectural drawing” (Line 414). Probably I did not run the program correctly, but this indicates a need for better documentation of the software.

Review Fig 1. The interface output from a run.

Review Fig 2. The interface output from another run.

For figures please see the attachment.

Reviewer #3: The PAAD is very similar to the idea proposed in the following paper, I would suggest explain the difference of your study from the published one.

https://www.researchgate.net/profile/Hexu-Liu/publication/333907195_Optimizing_multi-wall_panel_configuration_for_panelized_construction_using_BIM/links/5d0bcac5458515c11ceadb66/Optimizing-multi-wall-panel-configuration-for-panelized-construction-using-BIM.pdf

There are 4 algorithms included in the manuscript. And also, one Github link is provided. However, hard to connect the 4 algorithms with the Github Repo. There is no README file in the Github. Please make your data and scripts easy to follow.

Please add more details of explaining each figure. I only see the figure title.

Reviewer #4: Please see my comments/questions below:

Figures are not well labeled. When I read this manuscript, I had a hard time to figure out which figures are which.

The abbreviation 2DBP is introduced in line 36, so you can directly use it in line 154.

For equation (1) and (2), I would suggest to use left alignment.

I don’t understand equation (3), which does't look like a traditional math equation. Could you please explain it? For example, it seems that “fitness(P,W,V,O)” is a superscript of “P,W,O,V,M,C”, what does it mean? What are the subject of “minimize” and “subject to” in this equation?

6. PLOS authors have the option to publish the peer review history of their article (what does this mean?). If published, this will include your full peer review and any attached files.

Reviewer #1: No

Reviewer #2: No

Reviewer #3: No

Reviewer #4: No

---

## [Decision Letter · Decision Letter 1]

30 Apr 2024

PAAD: Panelization algorithm for architectural designs

PONE-D-24-12621R1

Dear Dr. Vijay Mago,

We’re pleased to inform you that your manuscript has been judged scientifically suitable for publication and will be formally accepted for publication once it meets all outstanding technical requirements.

Kind regards,

Chen Li, Ph.D.

Academic Editor

PLOS ONE

Additional Editor Comments (optional):

Reviewers' comments:

Reviewer's Responses to Questions

**Comments to the Author**

1. If the authors have adequately addressed your comments raised in a previous round of review and you feel that this manuscript is now acceptable for publication, you may indicate that here to bypass the “Comments to the Author” section, enter your conflict of interest statement in the “Confidential to Editor” section, and submit your "Accept" recommendation.

Reviewer #1: All comments have been addressed

Reviewer #2: All comments have been addressed

2. Is the manuscript technically sound, and do the data support the conclusions?

Reviewer #1: Yes

Reviewer #2: Yes

3. Has the statistical analysis been performed appropriately and rigorously? 

Reviewer #1: Yes

Reviewer #2: N/A

4. Have the authors made all data underlying the findings in their manuscript fully available?

Reviewer #1: Yes

Reviewer #2: Yes

5. Is the manuscript presented in an intelligible fashion and written in standard English?

Reviewer #1: Yes

Reviewer #2: Yes

6. Review Comments to the Author

Reviewer #1: I have no further comments on this version of research paper and am satisfied with the current improvement.

Reviewer #2: The authors added details to the figure legends, uploaded a binary file (JAR) of the software to GitHub, and revised the manuscript as well as the documentation. These improvements have adequately addressed my concerns and I don’t have further questions.

7. PLOS authors have the option to publish the peer review history of their article (what does this mean?). If published, this will include your full peer review and any attached files.

Reviewer #1: No

Reviewer #2: No

---

## [Editor Report · Acceptance letter]

15 May 2024

PONE-D-24-12621R1 

PLOS ONE

Dear Dr. Mago, 

I'm pleased to inform you that your manuscript has been deemed suitable for publication in PLOS ONE. Congratulations! Your manuscript is now being handed over to our production team.

Kind regards, 

on behalf of

Dr. Chen Li 

Academic Editor

PLOS ONE